# Multi-Fidelity Bayesian Optimization via Deep Neural Networks

**Shibo Li**
School of Computing
University of Utah
Salt Lake City, UT 84112
shibo@cs.utah.edu

**Wei Xing**
Scientific Computing and Imaging Institute
University of Utah
Salt Lake City, UT 84112
wxing@sci.utah.edu

**Robert M. Kirby**
School of Computing
University of Utah
Salt Lake City, UT 84112
kirby@cs.utah.edu

**Shandian Zhe**
School of Computing
University of Utah
Salt Lake City, UT 84112
zhe@cs.utah.edu

## Abstract

Bayesian optimization (BO) is a popular framework for optimizing black-box functions. In many applications, the objective function can be evaluated at multiple fidelities to enable a trade-off between the cost and accuracy. To reduce the optimization cost, many multi-fidelity BO methods have been proposed. Despite their success, these methods either ignore or over-simplify the strong, complex correlations across the fidelities. While the acquisition function is therefore easy and convenient to calculate, these methods can be inefficient in estimating the objective function. To address this issue, we propose Deep Neural Network Multi-Fidelity Bayesian Optimization (DNN-MFBO) that can flexibly capture all kinds of complicated relationships between the fidelities to improve the objective function estimation and hence the optimization performance. We use sequential, fidelity-wise Gauss-Hermite quadrature and moment-matching to compute a mutual information based acquisition function in a tractable and highly efficient way. We show the advantages of our method in both synthetic benchmark datasets and real-world applications in engineering design.

## 1   Introduction

Bayesian optimization (BO) (Mockus et al., 1978; Snoek et al., 2012) is a general and powerful approach for optimizing black-box functions. It uses a probabilistic surrogate model (typically Gaussian process (GP) (Rasmussen and Williams, 2006)) to estimate the objective function. By repeatedly maximizing an acquisition function computed with the information of the surrogate model, BO finds and queries at new input locations that are closer and closer to the optimum; meanwhile the new training examples are incorporated into the surrogate model to improve the objective estimation.

In practice, many applications allow us to query the objective function at different fidelities, where low fidelity queries are cheap yet inaccurate, and high fidelity queries more accurate but costly. For example, in physical simulation (Peherstorfer et al., 2018), the computation of an objective (*e.g.,* the elasticity of a part or energy of a system) often involves solving partial differential equations. Running a numerical solver with coarse meshes gives a quick yet rough result; using dense meshes substantially improves the accuracy but dramatically increases the computational cost. The multi-fidelity queries enable us to choose a trade-off between the cost and accuracy.

Accordingly, to reduce the optimization cost, many multi-fidelity BO methods (Huang et al., 2006; Lam et al., 2015; Kandasamy et al., 2016; Zhang et al., 2017; Takeno et al., 2019) have been proposed to jointly select the input locations and fidelities to best balance the optimization progress and query cost, *i.e.,* the benefit-cost ratio. Despite their success, these methods often ignore the strong, complex correlations between the function outputs at different fidelities, and learn an independent GP for each fidelity (Lam et al., 2015; Kandasamy et al., 2016). Recent works use multi-output GPs to capture the fidelity correlations. However, to avoid intractable computation of the acquisition function, they have to impose simplified correlation structures. For example, Takeno et al. (2019) assume a linear correlation between the fidelities; Zhang et al. (2017) use kernel convolution to construct the cross-covariance function, and have to choose simple, smooth kernels (*e.g.,* Gaussian) to ensure a tractable convolution. Therefore, the existing methods can be inefficient and inaccurate in estimating the objective function, which further lowers the optimization efficiency and increases the cost.

To address these issues, we propose DNN-MFBO, a deep neural network based multi-fidelity Bayesian optimization that is flexible enough to capture all kinds of complex (possibly highly nonlinear and nonstationary) relationships between the fidelities, and exploit these relationships to jointly estimate the objective function in all the fidelities to improve the optimization performance. Specifically, we stack a set of neural networks (NNs) where each NN models one fidelity. In each fidelity, we feed both the original input (to the objective) and output from the previous fidelity into the NN to propagate information throughout and to estimate the complex relationships across the fidelities. Then, the most challenging part is the calculation of the acquisition function. For efficient inference and tractable computation, we consider the NN weights in the output layer as random variables and all the other weights as hyper-parameters. We develop a stochastic variational learning algorithm to jointly estimate the posterior of the random weights and hyper-parameters. Next, we sequentially perform Gauss-Hermite quadrature and moment matching to approximate the posterior and conditional posterior of the output in each fidelity, based on which we calculate and optimize an information based acquisition function, which is not only computationally tractable and efficient, but also conducts maximum entropy search (Wang and Jegelka, 2017), the state-of-the-art criterion in BO.

For evaluation, we examined DNN-MFBO in three benchmark functions and two real-world applications in engineering design that requires physical simulations. The results consistently demonstrate that DNN-MFBO can optimize the objective function (in the highest fidelity) more effectively, meanwhile with smaller query cost, as compared with state-of-the-art multi-fidelity and single fidelity BO algorithms.

## 2    Background

**Bayesian optimization**. To optimize a black-box objective function $f : \mathcal{X} \to \mathbb{R}$, BO learns a probabilistic surrogate model to predict the function values across the input domain $\mathcal{X}$ and quantifies the uncertainty of the predictions. This information is used to calculate an acquisition function that measures the utility of querying at different input locations, which usually encodes a exploration-exploitation trade-off. By maximizing the acquisition function, BO finds new input locations at which to query, which are supposed to be closer to the optimum; meanwhile the new examples are added into the training set to improve the accuracy of the surrogate model. The most commonly used surrogate model is Gaussian process (GP) (Rasmussen and Williams, 2006). Given the training inputs $\mathbf{X} = [\mathbf{x}_1, \ldots, \mathbf{x}_N]^\top$ and (noisy) outputs $\mathbf{y} = [y_1, \ldots, y_N]^\top$, GP assumes the outputs follow a multivariate Gaussian distribution, $p(\mathbf{y}|\mathbf{X}) = \mathcal{N}(\mathbf{y}|\mathbf{m}, \mathbf{K} + \sigma^2\mathbf{I})$ where $\mathbf{m}$ are the values of the mean function at the inputs $\mathbf{X}$, $\mathbf{K}$ is a kernel matrix on $\mathbf{X}$, $[\mathbf{K}]_{ij} = k(\mathbf{x}_i, \mathbf{x}_j)$ ($k(\cdot, \cdot)$ is the kernel function), and $\sigma^2$ is the noise variance. The mean function is usually set to the constant function $0$ and so $\mathbf{m} = \mathbf{0}$. Due to the multi-variate Gaussian form, given a new input $\mathbf{x}^*$, the posterior distribution of the function output, $p(f(\mathbf{x}^*)|\mathbf{x}^*, \mathbf{X}, \mathbf{y})$ is a closed-form conditional Gaussian, and hence is convenient to quantify the uncertainty and calculate the acquisition function.

There are a variety of commonly used acquisition functions, such as expected improvement (EI) (Jones et al., 1998), upper confident bound (UCB) (Srinivas et al., 2010), entropy search (ES) (Hennig and Schuler, 2012), and predictive entropy search (PES) (Hernández-Lobato et al., 2014). A particularly successful recent addition is the max-value entropy search (MES) (Wang and Jegelka, 2017), which not only enjoys a global utility measure (like ES and PES), but also is computationally efficient (because it calculates the entropy of the function output rather than input like in ES/PES). Specifically, MES maximizes the mutual information between the function value and its maximum $f^*$ to find the

next input at which to query,

$$a(\mathbf{x}) = I\big(f(\mathbf{x}), f^*|\mathcal{D}\big) = H\big(f(\mathbf{x})|\mathcal{D}\big) - \mathbb{E}_{p(f^*|\mathcal{D})}[H\big(f(\mathbf{x})|f^*, \mathcal{D}\big)], \tag{1}$$

where $I(\cdot, \cdot)$ is the mutual information, $H(\cdot)$ the entropy, and $\mathcal{D}$ the training examples collected so far. Note that the function values and extremes are considered as generated from the posterior in the surrogate model, which includes all the knowledge we have for the black-box objective function.

**Multi-fidelity Bayesian optimization**. Many applications allow multi-fidelity queries of the objective function, $\{f_1(\mathbf{x}), \ldots, f_M(\mathbf{x})\}$, where the higher (larger) the fidelity $m$, the more accurate yet costly the query of $f_m(\cdot)$. Many studies have extended BO for multi-fidelity settings. For example, MF-GP-UCB (Kandasamy et al., 2016) starts from the lowest fidelity ($m = 1$), and queries the objective at each fidelity until the confidence band exceeds a particular threshold. Despite its effectiveness and theoretical guarantees, MF-GP-UCB learns an independent GP surrogate for each fidelity and ignores the strong correlations between the fidelities. Recent works use a multi-output GP to model the fidelity correlations. For example, MF-PES (Zhang et al., 2017) introduces a shared latent function, and uses kernel convolution to derive the cross-covariance between the fidelities. The most recent work, MF-MES (Takeno et al., 2019) introduces $C$ kernel functions $\{\kappa_c(\cdot, \cdot)\}$ and, for each fidelity $m$, $C$ latent features $\{\omega_{cm}\}$. The covariance function is defined as

$$k\big(f_m(\mathbf{x}), f_{m'}(\mathbf{x}')\big) = \sum_{c=1}^{C} (\omega_{cm}\omega_{cm'} + \tau_{cm}\delta_{mm'})\kappa_c(\mathbf{x}, \mathbf{x}'), \tag{2}$$

where $\tau_{cm} > 0$, $\delta_{mm'} = 1$ if and only if $m = m'$, and each kernel $\kappa_c(\cdot, \cdot)$ is usually assumed to be stationary, *e.g.*, Gaussian kernel.

# 3   Multi-Fidelity Modeling with Deep Neural Networks

Despite the success of existing multi-fidelity BO methods, they either overlook the strong, complex correlations between different fidelities (*e.g.*, MF-GP-UCB) or model these correlations with an over-simplified structure. For example, the convolved GP in MF-PES has to employ simple/smooth kernels (typically Gaussian) for both the latent function and convolution operation to obtain an analytical cross-covariance function, which has limited expressiveness. MF-MES essentially adopts a linear correlation assumption between the fidelities. According to (2), if we choose each $\kappa_c$ as a Gaussian kernel (with amplitude one), we have $k\big(f_m(\mathbf{x}), f_{m'}(\mathbf{x})\big) = \boldsymbol{\omega}_m^\top \boldsymbol{\omega}_{m'} + \delta_{mm'}\tau_m$ where $\boldsymbol{\omega}_m = [\omega_{1m}, \ldots, \omega_{Cm}]^\top$ and $\tau_m = \sum_{c=1}^{C} \tau_{cm}$. These correlation structures might be over-simplified and insufficient to estimate the complicated relationships between the fidelities (*e.g.*, highly nonlinear and nonstationary). Hence, they can limit the accuracy of the surrogate model and lower the optimization efficiency while increasing the query cost.

To address this issue, we use deep neural networks to build a multi-fidelity model that is flexible enough to capture all kinds of complicated relationships between the fidelities, taking advantage of the relationships to promote the accuracy of the surrogate model. Specifically, for each fidelity $m > 1$, we introduce a neural network (NN) parameterized by $\{\mathbf{w}_m, \boldsymbol{\theta}_m\}$, where $\mathbf{w}_m$ are the weights in the output layer and $\boldsymbol{\theta}_m$ the weights in all the other layers. Denote the NN input by $\mathbf{x}_m$, the output by $f_m(\mathbf{x})$ and the noisy observation by $y_m(\mathbf{x})$. The model is defined as

$$\mathbf{x}_m = [\mathbf{x}; f_{m-1}(\mathbf{x})], \quad f_m(\mathbf{x}) = \mathbf{w}_m^\top \boldsymbol{\phi}_{\boldsymbol{\theta}_m}(\mathbf{x}_m), \quad y_m(\mathbf{x}) = f_m(\mathbf{x}) + \epsilon_m, \tag{3}$$

where $\mathbf{x}$ is the original input to the objective function, $\boldsymbol{\phi}_{\boldsymbol{\theta}_m}(\mathbf{x}_m)$ is the output vector of the second last layer (hence parameterized by $\boldsymbol{\theta}_m$) which can be viewed as a set of nonlinear basis functions, and $\epsilon_m \sim \mathcal{N}(\epsilon_m|0, \sigma_m^2)$ is a Gaussian noise. The input $\mathbf{x}_m$ is obtained by appending the output from the previous fidelity to the original input. Through a series of linear and nonlinear transformations inside the NN, we obtain the output $f_m(\mathbf{x})$. In this way, we digest the information from the lower fidelities, and capture the complex relationships between the current and previous fidelities by learning a nonlinear mapping $f_m(\mathbf{x}) = h(\mathbf{x}, f_{m-1}(\mathbf{x}))$, where $h(\cdot)$ is fulfilled by the NN. When $m = 1$, we set $\mathbf{x}_m = \mathbf{x}$. A graphical representation of our model is given in Fig. 1 of the supplementary material.

We assign a standard normal prior over each $\mathbf{w}_m$. Following (Snoek et al., 2015), we consider all the remaining NN parameters as hyper-parameters. Given the training set $\mathcal{D} = \{\{(\mathbf{x}_{nm}, y_{nm})\}_{n=1}^{N_m}\}_{m=1}^{M}$, the joint probability of our model is

$$p(\mathcal{W}, \mathcal{Y}|\mathcal{X}, \Theta, \mathbf{s}) = \prod_{m=1}^{M} \mathcal{N}(\mathbf{w}_m|\mathbf{0}, \mathbf{I}) \prod_{n=1}^{N_m} \mathcal{N}\big(y_{nm}|f_m(\mathbf{x}_{nm}), \sigma_m^2\big), \tag{4}$$

where $\mathcal{W} = \{\mathbf{w}_m\}$, $\Theta = \{\boldsymbol{\theta}_m\}$, $\mathbf{s} = [\sigma_1^2, \ldots, \sigma_M^2]^\top$, and $\mathcal{X}$, $\mathcal{Y}$ are the inputs and outputs in $\mathcal{D}$.

In order to obtain the posterior distribution of our model (which is in turn used to compute the acquisition function), we develop a stochastic variational learning algorithm. Specifically, for each $\mathbf{w}_m$, we introduce a multivariate Gaussian posterior, $q(\mathbf{w}_m) = \mathcal{N}(\mathbf{w}_m | \boldsymbol{\mu}_m, \boldsymbol{\Sigma}_m)$. We further parameterize $\boldsymbol{\Sigma}_m$ with its Cholesky decomposition to ensure the positive definiteness, $\boldsymbol{\Sigma}_m = \mathbf{L}_m \mathbf{L}_m^\top$ where $\mathbf{L}_m$ is a lower triangular matrix. We assume $q(\mathcal{W}) = \prod_{m=1}^M q(\mathbf{w}_m)$, and construct a variational model evidence lower bound (ELBO), $\mathcal{L}(q(\mathcal{W}), \Theta, \mathbf{s}) = \mathbb{E}_q[\log(p(\mathcal{W}, \mathcal{Y} | \mathcal{X}, \Theta, \mathbf{s})/q(\mathcal{W}))]$. We then maximize the ELBO to jointly estimate the variational posterior $q(\mathcal{W})$ and all the other hyper-parameters. The ELBO is analytically intractable, and we use the reparameterization trick (Kingma and Welling, 2013) to conduct efficient stochastic optimization. The details are given in the supplementary material (Sec. 3).

# 4  Multi-Fidelity Optimization with Max-Value Entropy Search

We now consider an acquisition function to select both the fidelities and input locations at which we query during optimization. Following (Takeno et al., 2019), we define the acquisition function as

$$a(\mathbf{x}, m) = \frac{1}{\lambda_m} I(f^*, f_m(\mathbf{x}) | \mathcal{D}) = \frac{1}{\lambda_m} \left( H(f_m(\mathbf{x}) | \mathcal{D}) - \mathbb{E}_{p(f^* | \mathcal{D})} \left[ H(f_m(\mathbf{x}) | f^*, \mathcal{D}) \right] \right) \quad (5)$$

where $\lambda_m > 0$ is the cost of querying with fidelity $m$. In each step, we maximize the acquisition function to find a pair of input location and fidelity that provides the largest benefit-cost ratio.

However, given the model inference result, *i.e.,* $p(\mathcal{W} | \mathcal{D}) \approx q(\mathcal{W})$, a critical challenge is to compute the posterior distribution of the output in each fidelity, $p(f_m(\mathbf{x}) | \mathcal{D})$, and use them to compute the acquisition function. Due to the nonlinear coupling of the outputs in different fidelities (see (3)), the computation is analytically intractable. To address this issue, we conduct fidelity-wise moment matching and Gauss-Hermite quadrature to approximate each $p(f_m(\mathbf{x}) | \mathcal{D})$ as a Gaussian distribution.

## 4.1  Computing Output Posteriors

Specifically, we first assume that we have obtained the posterior of the output for fidelity $m-1$, $p(f_{m-1}(\mathbf{x}) | \mathcal{D}) \approx \mathcal{N}(f_{m-1} | \alpha_{m-1}(\mathbf{x}), \eta_{m-1}(\mathbf{x}))$. For convenience, we slightly abuse the notation and use $f_{m-1}$ and $f_m$ to denote $f_{m-1}(\mathbf{x})$ and $f_m(\mathbf{x})$, respectively. Now we consider calculating $p(f_m | \mathcal{D})$. According to (3), we have $f_m = \mathbf{w}_m^\top \boldsymbol{\phi}_{\boldsymbol{\theta}_m}([\mathbf{x}; f_{m-1}])$. Based on our variational posterior $q(\mathbf{w}_m) = \mathcal{N}(\mathbf{w}_m | \boldsymbol{\mu}_m, \mathbf{L}_m \mathbf{L}_m^\top)$, we can immediately derive the conditional posterior $p(f_m | f_{m-1}, \mathcal{D}) = \mathcal{N}(f_m | u(f_{m-1}, \mathbf{x}), \gamma(f_{m-1}, \mathbf{x}))$ where $u(f_{m-1}, \mathbf{x}) = \boldsymbol{\mu}_m^\top \boldsymbol{\phi}_{\boldsymbol{\theta}_m}([\mathbf{x}; f_{m-1}])$ and $\gamma(f_{m-1}, \mathbf{x}) = \|\mathbf{L}_m^\top \boldsymbol{\phi}_{\boldsymbol{\theta}_m}([\mathbf{x}; f_{m-1}])\|^2$. Here $\|\cdot\|^2$ is the square norm. We can thereby read out the first and second conditional moments,

$$\mathbb{E}[f_m | f_{m-1}, \mathcal{D}] = u(f_{m-1}, \mathbf{x}), \quad \mathbb{E}[f_m^2 | f_{m-1}, \mathcal{D}] = \gamma(f_{m-1}, \mathbf{x}) + u(f_{m-1}, \mathbf{x})^2. \quad (6)$$

To obtain the moments, we need to take the expectation of the conditional moments w.r.t $p(f_{m-1} | \mathcal{D}) \approx \mathcal{N}(f_{m-1} | \alpha_{m-1}(\mathbf{x}), \eta_{m-1}(\mathbf{x}))$. While the conditional moments are nonlinear to $f_{m-1}$ and their expectation is not analytical, we can use Gauss-Hermite quadrature to give an accurate, closed-form approximation,

$$\mathbb{E}[f_m | \mathcal{D}] = \mathbb{E}_{p(f_{m-1} | \mathcal{D})} \mathbb{E}[f_m | f_{m-1}, \mathcal{D}] \approx \sum_k g_k \cdot u(t_k, \mathbf{x}),$$

$$\mathbb{E}[f_m^2 | \mathcal{D}] = \mathbb{E}_{p(f_{m-1} | \mathcal{D})} \mathbb{E}[f_m^2 | f_{m-1}, \mathcal{D}] \approx \sum_k g_k \cdot [\gamma(t_k, \mathbf{x}) + u(t_k, \mathbf{x})^2], \quad (7)$$

where $\{g_k\}$ and $\{t_k\}$ are quadrature weights and nodes, respectively. Note that each node $t_k$ is determined by $\alpha_{m-1}(\mathbf{x})$ and $\eta_{m-1}(\mathbf{x})$. We then use these moments to construct a Gaussian posterior approximation, $p(f_m | \mathcal{D}) \approx \mathcal{N}(f_m | \alpha_m(\mathbf{x}), \eta_m(\mathbf{x}))$ where $\alpha_m(\mathbf{x}) = \mathbb{E}[f_m | \mathcal{D}]$ and $\eta_m(\mathbf{x}) = \mathbb{E}[f_m^2 | \mathcal{D}] - \mathbb{E}[f_m | \mathcal{D}]^2$. This is called moment matching, which is widely used and very successful in approximate Bayesian inference, such as expectation-propagation (Minka, 2001). One may concern if the quadrature will give a positive variance. This is guaranteed by the follow lemma.

**Lemma 4.1.** *As long as the conditional posterior variance $\gamma(f_{m-1}, \mathbf{x}) > 0$, the posterior variance $\eta_m(\mathbf{x})$, computed based on the quadrature in (7), is positive.*

The proof is given in the supplementary material. Following the same procedure, we can compute the posterior of the output in fidelity $m + 1$. Note that when $m = 1$, we do not need quadrature because the input of the NN is the same as the original input, not including other NN outputs. Hence, we can derive the Gaussian posterior outright from $q(\mathbf{w}_1)$ — $p(f_1(\mathbf{x})|\mathcal{D}) = \mathcal{N}(f_1(\mathbf{x})|\alpha_1(\mathbf{x}), \eta_1(\mathbf{x}))$, where $\alpha_1(\mathbf{x}) = \boldsymbol{\mu}_1^\top \boldsymbol{\phi}_{\boldsymbol{\theta}_1}(\mathbf{x})$ and $\eta_1(\mathbf{x}) = \|\mathbf{L}_1^\top \boldsymbol{\phi}_{\boldsymbol{\theta}_1}(\mathbf{x})\|^2$.

## 4.2 Computing Acquisition Function

Given the posterior of the NN output in each fidelity, $p(f_m(\mathbf{x})|\mathcal{D}) \approx \mathcal{N}(f_m(\mathbf{x})|\alpha_m(\mathbf{x}), \eta_m(\mathbf{x}))(1 \leq m \leq M)$, we consider how to compute the acquisition function (5). Due to the Gaussian posterior, the first entropy term is straightforward, $H(f^m(\mathbf{x})|\mathcal{D}) = \frac{1}{2} \log (2\pi e \eta_m(\mathbf{x}))$. The second term — a conditional entropy, however, is intractable. Hence, we follow (Wang and Jegelka, 2017) to use a Monte-Carlo approximation,

$$\mathbb{E}_{p(f^*|\mathcal{D})}[H(f^m(\mathbf{x})|f^*, \mathcal{D})] \approx \frac{1}{|\mathcal{F}|} \sum_{f^* \in \mathcal{F}^*} H(f_m(\mathbf{x})|f^*, \mathcal{D}),$$

where $\mathcal{F}^*$ are a collection of independent samples of the function maximums based on the posterior distribution of our model. To obtain a sample of the function maximum, we first generate a posterior sample for each $\mathbf{w}_m$, according to $q(\mathbf{w}_m) = \mathcal{N}(\mathbf{w}_m|\boldsymbol{\mu}_m, \mathbf{L}_m \mathbf{L}_m^\top)$. We replace each $\mathbf{w}_m$ by their sample in calculating $f^M(\mathbf{x})$ so as to obtain a posterior sample of the objective function. We then maximize this sample function to obtain one instance of $f^*$. We use L-BFGS (Liu and Nocedal, 1989) for optimization.

Given $f^*$, the computation of $H(f_m(\mathbf{x})|f^*, \mathcal{D}) = H(f_m(\mathbf{x})|\max f_M(\mathbf{x}) = f^*, \mathcal{D})$ is still intractable. We then follow (Wang and Jegelka, 2017) to calculate $H(f_m(\mathbf{x})|f_M(\mathbf{x}) \leq f^*, \mathcal{D})$ instead as a reasonable approximation. For $m = M$, the entropy is based on a truncated Gaussian distribution, $p(f_M(\mathbf{x})|f_M(\mathbf{x}) \leq f^*, \mathcal{D}) \propto \mathcal{N}(f_M(\mathbf{x})|\alpha_M(\mathbf{x}), \eta_M(\mathbf{x})) \mathbb{1}(f_M(\mathbf{x}) \leq f^*)$ where $\mathbb{1}(\cdot)$ is the indicator function, and is given by

$$H(f_m(\mathbf{x})|f_M(\mathbf{x}) \leq f^*, \mathcal{D}) = \log (\sqrt{2\pi e \eta_M(\mathbf{x})}\Phi(\beta)) - \beta \cdot \mathcal{N}(\beta|0, 1)/(2\Phi(\beta)), \quad (8)$$

where $\Phi(\cdot)$ is the cumulative density function (CDF) of the standard normal distribution, and $\beta = (f^* - \alpha_M(\mathbf{x}))/\sqrt{\eta_M(\mathbf{x})}$. When $m < M$, the entropy is based on the conditional distribution

$$p(f_m(\mathbf{x})|f_M(\mathbf{x}) \leq f^*, \mathcal{D}) = \frac{1}{Z} \cdot p(f_m(\mathbf{x})|\mathcal{D})p(f_M(\mathbf{x}) \leq f^*|f_m(\mathbf{x}), \mathcal{D})$$

$$\approx \frac{1}{Z} \cdot \mathcal{N}(f_m(\mathbf{x})|\alpha_m(\mathbf{x}), \eta_m(\mathbf{x}))p(f_M(\mathbf{x}) \leq f^*|f_m(\mathbf{x}), \mathcal{D}). \quad (9)$$

where $Z$ is the normalizer. To obtain $p(f_M(\mathbf{x}) \leq f^*|f_m(\mathbf{x}), \mathcal{D})$, we first consider how to compute $p(f_M(\mathbf{x})|f_m(\mathbf{x}), \mathcal{D})$. According to (3), it is trivial to derive that

$$p(f_{m+1}(\mathbf{x})|f_m(\mathbf{x}), \mathcal{D}) = \mathcal{N}(f_{m+1}|\widehat{\alpha}_{m+1}(\mathbf{x}, f_m), \widehat{\eta}_{m+1}(\mathbf{x}, f_m)),$$

where $\widehat{\alpha}_{m+1}(\mathbf{x}, f_m) = \boldsymbol{\mu}_{m+1}^\top \boldsymbol{\phi}_{\boldsymbol{\theta}_{m+1}}([\mathbf{x}; f_m])$ and $\widehat{\eta}_{m+1}(\mathbf{x}, f_m) = \|\mathbf{L}_{m+1}^\top \boldsymbol{\phi}_{\boldsymbol{\theta}_{m+1}}([\mathbf{x}; f_m])\|^2$. Note that we again use $f_{m+1}$ and $f_m$ to denote $f_{m+1}(\mathbf{x})$ and $f_m(\mathbf{x})$ for convenience. Next, we follow the same method as in Section 4.1 to sequentially obtain the conditional posterior for each higher fidelity, $p(f_{m+k}|f_m, \mathcal{D})(1 < k \leq M - m)$. In more detail, we first base on $q(\mathbf{w}_{m+k})$ to derive the conditional moments $\mathbb{E}(f_{m+k}|f_{m+k-1}, f_m, \mathcal{D})$ and $\mathbb{E}(f_{m+k}^2|f_{m+k-1}, f_m, \mathcal{D})$. They are calculated in the same way as in (6), because $f_{m+k}$ are independent to $f_m$ conditioned on $f_{m+k-1}$. Then we take the expectation of the conditional moments w.r.t $p(f_{m+k-1}|f_m, \mathcal{D})$ (that is Gaussian) to obtain $\mathbb{E}(f_{m+k}|f_m, \mathcal{D})$ and $\mathbb{E}(f_{m+k}^2|f_m, \mathcal{D})$. This again can be done by Gauss-Hermite quadrature. Finally, we use these moments to construct a Gaussian approximation to the conditional posterior,

$$p(f_{m+k}|f_m, \mathcal{D}) \approx \mathcal{N}(f_{m+k}|\widehat{\alpha}_{m+k}(\mathbf{x}, f_m), \widehat{\eta}_{m+k}(\mathbf{x}, f_m)), \quad (10)$$

where $\widehat{\alpha}_{m+k}(\mathbf{x}, f_m) = \mathbb{E}(f_{m+k}|f_m, \mathcal{D})$ and $\widehat{\eta}_{m+k}(\mathbf{x}, f_m) = \mathbb{E}(f_{m+k}^2|f_m, \mathcal{D}) - \mathbb{E}(f_{m+k}|f_m, \mathcal{D})^2$. According to Lemma 4.1, we guarantee $\widehat{\eta}_{m+k}(\mathbf{x}, f_m) > 0$. Now we can obtain

$$p(f_m(\mathbf{x})|f_M(\mathbf{x}) \leq f^*, \mathcal{D}) \approx \frac{1}{Z} \cdot \mathcal{N}(f_m|\alpha_m(\mathbf{x}), \eta_m(\mathbf{x}))\Phi\big(\frac{f^* - \widehat{\alpha}_M(\mathbf{x}, f_m)}{\sqrt{\widehat{\eta}_M(\mathbf{x}, f_m)}}\big). \quad (11)$$

In order to compute the entropy analytically, we use moment matching again to approximate this distribution as a Gaussian distribution. To this end, we use Gauss-Hermite quadrature to compute three integrals, $Z = \int R(f_m) \cdot \mathcal{N}\big(f_m|\alpha_m(\mathbf{x}), \eta_m(\mathbf{x})\big)\mathrm{d}f_m$, $Z_1 = \int f_m R(f_m) \cdot \mathcal{N}\big(f_m|\alpha_m(\mathbf{x}), \eta_m(\mathbf{x})\big)\mathrm{d}f_m$, and $Z_2 = \int f_m^2 R(f_m) \cdot \mathcal{N}\big(f_m|\alpha_m(\mathbf{x}), \eta_m(\mathbf{x})\big)\mathrm{d}f_m$, where $R(f_m) = \Phi\big((f^* - \widehat{\alpha}_M(\mathbf{x}, f_m))/\sqrt{\widehat{\eta}_M(\mathbf{x}, f_m)}\big)$. Then we can obtain $\mathbb{E}[f_m|f_M \leq f^*, \mathcal{D}] = Z1/Z$ and $\mathbb{E}[f_m^2|f_M \leq f^*, \mathcal{D}] = Z_2/Z$, based on which we approximate

$$p(f_m(\mathbf{x})|f_M(\mathbf{x}) \leq f^*, \mathcal{D}) \approx \mathcal{N}\big(f_m|Z_1/Z, Z_2/Z - Z_1^2/Z^2\big). \tag{12}$$

Following the same idea to prove Lemma 4.1, we can show that the variance is non-negative. See the details in the supplementary material (Sec. 5). With the Gaussian form, we can analytically compute the entropy, $H\big(f_m(\mathbf{x})|f_M(\mathbf{x}) \leq f^*, \mathcal{D}\big) = \frac{1}{2}\log\big(2\pi e(Z_2/Z - Z_1^2/Z^2)\big)$.

Although our calculation of the acquisition function is quite complex, due to the analytical form, we can use automatic differentiation libraries (Baydin et al., 2017), to compute the gradient efficiently and robustly for optimization. In our experiments, we used TensorFlow (Abadi et al., 2016) and L-BFGS to maximize the acquisition function to find the fidelity and input location we query at in the next step. Our multi-fidelity Bayesian optimization algorithm is summarized in Algorithm 1.

---

**Algorithm 1** DNN-MFBO ($\mathcal{D}, M, T, \{\lambda_m\}_{m=1}^M$ )

---

1: Learn the DNN-based multi-fidelity model (4) on $\mathcal{D}$ with stochastic variational learning.
2: **for** $t = 1, \ldots, T$ **do**
3:     Generate $\mathcal{F}^*$ from the variational posterior $q(\mathcal{W})$ and the NN output at fidelity $M$, *i.e.,* $f_M(\mathbf{x})$
4:     $(\mathbf{x}_t, m_t) = \text{argmax}_{\mathbf{x} \in \mathcal{X}, 1 \leq m \leq M} \text{MutualInfo}(\mathbf{x}, m, \lambda_m, \mathcal{F}^*, \mathcal{D}, M)$
5:     $\mathcal{D} \leftarrow \mathcal{D} \cup \{(\mathbf{x}_t, m_t)\}$
6:     Re-train the DNN-based multi-fidelity model on $\mathcal{D}$
7: **end for**

---

---

**Algorithm 2** MutualInfo($\mathbf{x}, m, \lambda_m, \mathcal{F}^*, \mathcal{D}, M$)

---

1: Compute each $p(f_m(\mathbf{x})|\mathcal{D}) \approx \mathcal{N}\big(f_m|\alpha_m(\mathbf{x}), \eta_m(\mathbf{x})\big)$ (Sec. 4.1)
2: $H_0 \leftarrow \frac{1}{2}\log(2\pi e \eta_m(\mathbf{x}))$, $H_1 \leftarrow 0$
3: **for** $f^* \in \mathcal{F}^*$ **do**
4:     **if** $m = M$ **then**
5:         Use (8) to compute $H(f_m|f_M \leq f^*, \mathcal{D})$ and add it to $H_1$
6:     **else**
7:         Compute $p(f_m(\mathbf{x})|f_M(\mathbf{x}), \mathcal{D})$ following (10) and $p(f_m(\mathbf{x})|f_M(\mathbf{x}) \leq f^*, \mathcal{D})$ with (12)
8:         $H_1 \leftarrow H_1 + \frac{1}{2}\log\big(2\pi e(Z_2/Z - Z_1^2/Z^2)\big)$
9:     **end if**
10: **end for**
11: **return** $(H_0 - H_1/|\mathcal{F}^*|)/\lambda_m$

---

## 5  Related Work

Most surrogate models used in Bayesian optimization (BO) (Mockus, 2012; Snoek et al., 2012) are based on Gaussian processes (GPs) (Rasmussen and Williams, 2006), partly because their closed-form posteriors (Gaussian) are convenient to quantify the uncertainty and calculate the acquisition functions. However, GPs are known to be costly for training, and the exact inference takes $\mathcal{O}(N^3)$ time complexity ($N$ is the number of samples). Recently, Snoek et al. (2015) showed deep neural networks (NNs) can also be used in BO and performs very well. The training of NNs are much more efficient ($\mathcal{O}(N)$). To conveniently quantify the uncertainty, Snoek et al. (2015) consider the NN weights in the output layer as random variables and all the other weights as hyper-parameters (like the kernel parameters in GPs). They first obtain a point estimation of the hyper-parameters (typically through stochastic training). Then they fix the hyper-parameters and compute the posterior distribution of the random weights (in the last layer) and NN output — this can be viewed as the inference for Bayesian linear regression. In our multi-fidelity model, we also only consider the NN weights in the output layer of each fidelity as random variables. However, we jointly estimate the hyper-parameters and posterior distribution of the random weights. Since the NN outputs in successive fidelities are coupled non-linearly, we use the variational estimation framework (Wainwright et al., 2008).

Many multi-fidelity BO algorithms have been proposed. For example, Huang et al. (2006); Lam et al. (2015); Picheny et al. (2013) augmented the standard EI for the multi-fidelity settings. Kandasamy et al. (2016, 2017) extended GP upper confidence bound (GP-UCB) (Srinivas et al., 2010). Poloczek et al. (2017); Wu and Frazier (2017) developed multi-fidelity BO with knowledge gradients (Frazier et al., 2008). EI is a local measure of the utility and UCB requires us to explicitly tune the exploit-exploration trade-off. The recent works also extend the information-based acquisition functions to enjoy a global utility for multi-fidelity optimization, *e.g.,* (Swersky et al., 2013; Klein et al., 2017) using entropy search (ES), (Zhang et al., 2017; McLeod et al., 2017) (PES) using predictive entropy search (PES), and (Song et al., 2019; Takeno et al., 2019) using max-value entropy search (MES). Note that ES and PES are computationally more expensive than MES because the former calculate the entropy of the input (vector) and latter the output scalar. Despite the great success of the existing methods, they either ignore or oversimplify the complex correlations across the fidelities, and hence might hurt the accuracy of the surrogate model and further the optimization performance. For example, Picheny et al. (2013); Lam et al. (2015); Kandasamy et al. (2016); Poloczek et al. (2017) train an independent GP for each fidelity; Song et al. (2019) combined all the examples indiscriminately to train a single GP; Huang et al. (2006); Takeno et al. (2019) assume a linear correlation structure between fidelities, and Zhang et al. (2017) used the convolution operation to construct the covariance and so the involved kernels have to be simple and smooth enough (yet less expressive) to obtain an analytical form. To overcome these limitations, we propose an NN-based multi-fidelity model, which is flexible enough to capture arbitrarily complex relationships between the fidelities and to promote the performance of the surrogate model. Recently, a NN-based multi-task model (Perrone et al., 2018) was also developed for BO and hyper-parameter transfer learning. The model uses an NN to construct a shared feature map (*i.e.,* bases) across the tasks, and generates the output of each task by a linear combination of the latent features. While this model can also be used for multi-fidelity BO (each task corresponds to one fidelity), it views each fidelity as symmetric and does not reflect the monotonicity of function accuracy/importance along with the fidelities. More important, the model does not capture the correlation between fidelities — given the shared bases, different fidelities are assumed to be independent. Finally, while a few algorithms deal with continuous fidelities, *e.g.,* (Kandasamy et al., 2017; McLeod et al., 2017; Wu and Frazier, 2017), we focus on discrete fidelities in this work.

# 6 Experiment

## 6.1 Synthetic Benchmarks

We first evaluated DNN-MFBO in three popular synthetic benchmark tasks. (1) *Branin* function (Forrester et al., 2008; Perdikaris et al., 2017) with three fidelities. The input is two dimensional and ranges from $[-5, 10] \times [0, 15]$. (2) *Park1* function (Park, 1991) with two fidelities. The input is four dimensional and each dimension is in $[0, 1]$. (3) *Levy* function (Laguna and Martí, 2005), having three fidelities and two dimensional inputs. The domain is $[-10, 10] \times [-10, 10]$. For each objective function, between fidelities can be nonlinear and/or nonstationary transformations. The detailed definitions are given in the supplementary material (Sec. 1).

**Competing Methods.** We compared with the following popular and state-of-the-art multi-fidelity BO algorithms: (1) Multi-Fidelity Sequential Kriging (MF-SKO) (Huang et al., 2006) that models the function of the current fidelity as the function of the previous fidelity plus a GP, (2) MF-GP-UCB (Kandasamy et al., 2016), (3) Multi-Fidelity Predictive Entropy Search (MF-PES) (Zhang et al., 2017) and (4) Multi-Fidelity Maximum Entropy Search (MF-MES) (Takeno et al., 2019). These algorithms extend the standard BO with EI, UCB, PES and MES principles respectively. We also compared with (5) multi-task NN based BO (MTNN-BO) by Perrone et al. (2018), where a set of latent bases (generated by an NN) are shared across the tasks, and the output of each task (*i.e.,* fidelity) is predicted by a linear combination of the bases. We tested the single fidelity BO with MES, named as (5) SF-MES (Wang and Jegelka, 2017). SF-MES only queries the objective at the highest fidelity.

**Settings and Results.** We implemented our method and MTNN-BO with TensorFlow. We used the original Matlab implementation for MF-GP-UCB (https://github.com/kirthevasank/mf-gp-ucb), MF-PES (https://github.com/YehongZ/MixedTypeBO) and SF-MES (https://github.com/zi-w/Max-value-Entropy-Search/), and Python/Numpy implementation for MF-MES. MF-SKO was implemented with Python as well. SF-MES and MF-GP-UCB used the Squared Exponential (SE) kernel. MF-PES used the Automatic Relevance Determination (ARD) kernel. MF-MES and MF-SKO used the Radial

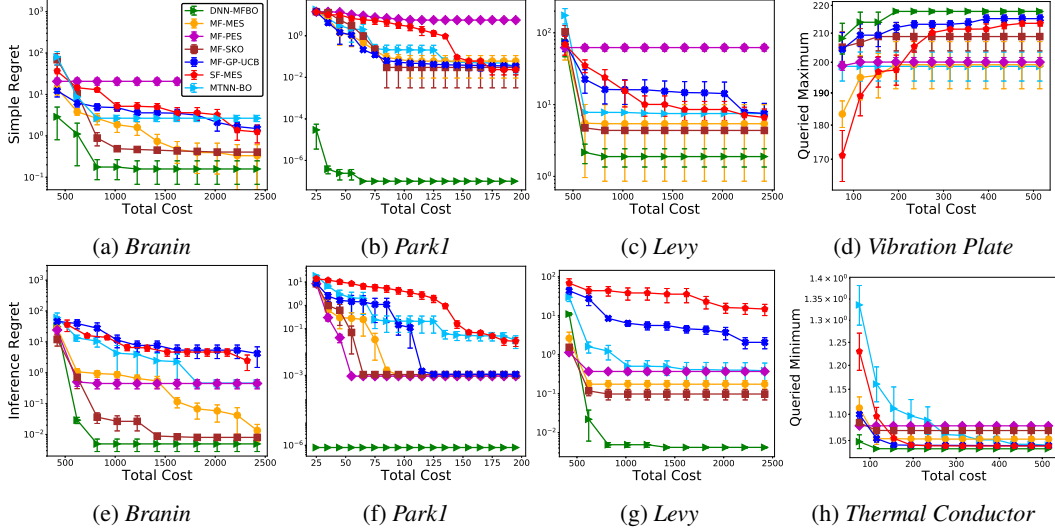

Figure 1: Simple and Inference regrets on three synthetic benchmark tasks (a-c, e-g) and the optimum queried function values (d, h) along with the query cost.

Basis (RBF) kernel (within each fidelity). We used the default settings in their implementations. For DNN-MFBO and MTNN-Bo, we used ReLU activation. To identify the architecture of the neural network in each fidelity and learning rate, we first ran the AutoML tool SMAC3 (https://github.com/automl/SMAC3) on the initial training dataset (we randomly split the data into half for training and the other half for test, and repeated multiple times to obtain a cross-validation accuracy to guide the search) and then manually tuned these hyper-parameters. The depth and width of each network were chosen from $[2, 12]$ and $[32, 512]$, and the learning rate $[10^{-5}, 10^{-1}]$. We used ADAM (Kingma and Ba, 2014) for stochastic training. The number of epochs was set to $5,000$, which is enough for convergence. To optimize the acquisition function, MF-MES and MF-PES first run a global optimization algorithm DIRECT (Jones et al., 1993; Gablonsky et al., 2001) and then use the results as the initialization to run L-BFGS. SF-MES uses a grid search first and then runs L-BFGS. DNN-MFBO and MTNN-BO directly use L-BFGS with a random initialization. To obtain the initial training points, we randomly query in each fidelity. For *Branin* and *Levy*, we generated 20, 20 and 2 training samples for the first, second and third fidelity, respectively. For *Park1*, we generated 5 and 2 examples for the first and second fidelity. The query costs is $(\lambda_1, \lambda_2, \lambda_3) = (1, 10, 100)$. We examined the simple regret (SR) and inference regret (IR). SR is defined as the difference between the global optimum and the best queried function value so far: $\max_{\mathbf{x} \in \mathcal{X}} f_M(\mathbf{x}) - \max_{i \in \{i | i \in [t], m_i = M\}} f_M(\mathbf{x}_i)$; IR is the difference between the global optimum and the optimum estimated by the surrogate model: $\max_{\mathbf{x} \in \mathcal{X}} f_M(\mathbf{x}) - \max_{\mathbf{x} \in \mathcal{X}} \widehat{f}_M(\mathbf{x})$ where $\widehat{f}_M(\cdot)$ is the estimated objective. We repeated the experiment for five times, and report on average how the simple and inference regrets vary along with the query cost in Fig. 1 (a-c, e-g). We also show the standard error bars. As we can see, in all the three tasks, DNN-MFBO achieves the best regrets with much smaller or comparable querying costs. The best regrets obtained by our method are much smaller (often orders of magnitude) than the baselines. In particular, DNN-MFBO almost achieved the global optimum after querying one point (IR $< 10^{-6}$) (Fig. 1f). These results demonstrate our DNN based surrogate model is more accurate in estimating the objective. Furthermore, our method spends less or comparable cost to achieve the best regrets, showing a much better benefit/cost ratio.

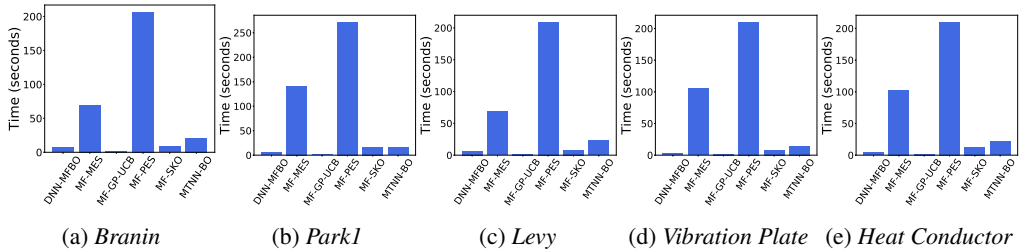

Figure 2: The average query time on three synthetic tasks (a-c) and two real-world applications (d-e).

### 6.2 Real-World Applications in Engineering Design

**Mechanical Plate Vibration Design.** We aim to optimize three material properties, Young's modulus (in $[1 \times 10^{11}, 5 \times 10^{11}]$), Poisson's ratio (in $[0.2, 0.6]$) and mass density (in $[6 \times 10^3, 9 \times 10^3]$), to maximize the fourth vibration mode frequency of a 3-D simply supported, square, elastic plate, of size $10 \times 10 \times 1$. To evaluate the frequency, we need to run a numerical solver on the discretized plate. We considered two fidelities, one with a coarse mesh and the other a dense mesh. The details about the settings of the solvers are provided the supplementary document.

**Thermal Conductor Design.** Given the property of a particular thermal conductor, our goal is to optimize the shape of the central hole where we install/fix the conductor to make the heat conduction (from left to right) to be as as fast as possible. The shape of the hole (an ellipse) is described by three parameters: x-radius, y-radius and angle. We used the time to reach 70 degrees as the objective function value and we want to minimize the objective. We need to run numerical solvers to calculate the objective. We considered two fidelities. The details are given in the supplementary material.

For both problems, we randomly queried at 20 and 5 inputs in the low and high fidelities respectively, at the beginning. The query cost is $(\lambda_1, \lambda_2) = (1, 10)$. We then ran each algorithm until convergence. We repeated the experiments for five times. Since we do not know the ground-truth of the global optimum, we report how the average of the best function values queried improves along with the cost. The results are shown in Fig. 1d and h. As we can see, in both applications, DNN-MFBO reaches the maximum/minimum function values with a smaller cost than all the competing methods, which is consistent with results in the synthetic benchmark tasks.

Finally, we examined the average query time of each multi-fidelity BO method, which is spent in calculating and optimizing the acquisition function to find new inputs and fidelities to query at in each step. For a fair comparison, we ran all the methods on a Linux workstation with a 16-core Intel(R) Xeon(R) CPU E5-2670 and 16GB RAM. As shown in Fig. 2, DNN-MFBO spends much less time than MF-MES and MF-PES that are based on multi-output GPs, and the speed of DNN-MFBO is close or comparable to MF-GP-UCB and MF-SKO, which use independent and additive GPs for each fidelity, respectively. On average, DNN-MFBO achieves 25x and 60x speedup over MF-MES and MF-PES. One reason might be that DNN-MFBO simply adopts a random initialization for L-BFGS rather than runs an expensive global optimization (so does MTNN-BO). However, as we can see from Fig. 1, DNN-MFBO still obtains new input and fidelities that achieve much better benefit/cost ratio. On the other hand, the close speed to MF-GP-UCB and MF-SKO also demonstrate that our method is efficient in acquisition function calculation, despite its seemingly complex approximations.

## 7 Conclusion

We have presented DNN-MFBO, a deep neural network based multi-fidelity Bayesian optimization algorithm. Our DNN surrogate model is flexible enough to capture the strong and complicated relationships between fidelities and promote objective estimation. Our information based acquisition function not only enjoys a global utility measure, but also is computationally tractable and efficient.

### Acknowledgments

This work has been supported by DARPA TRADES Award HR0011-17-2-0016 and NSF IIS-1910983.

### Broader Impact

This work can be used in a variety of engineering design problems that involve intensive computation, *e.g.,* finite elements or differences. Hence, the work has potential positive impacts in the society if it is used to design passenger aircrafts, biomedical devices, automobiles, and all the other devices or machines that can benefit human lives. At the same time, this work may have some negative consequences if it is used to design weapons or weapon parts.

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
