[Supplementary Material]

## Supplementary Material

Figure 1: Graphical representation of the DNN based multi-fidelity surrogate model. The output in each fidelity $f_m(\mathbf{x})$ $(1 \le m \le M)$ is fulfilled by a (deep) neural network.

# 1 Definitions of Synthetic Benchmark Functions

In the experiments, we used three synthetic benchmark tasks to evaluate our method. The definitions of the objective functions are given as follows.

## 1.1 Branin Function

The input is two dimensional, $\mathbf{x} = [x_1, x_2] \in [-5, 10] \times [0, 15]$. We have three fidelities to query the function, which, from high to low, are given by

$$
\begin{aligned}
f_3(\mathbf{x}) &= -\left(\frac{-1.275x_1^2}{\pi^2} + \frac{5x_1}{\pi} + x_2 - 6\right)^2 - \left(10 - \frac{5}{4\pi}\right)\cos(x_1) - 10, \\
f_2(\mathbf{x}) &= -10\sqrt{-f_3(x-2)} - 2(x_1 - 0.5) + 3(3x_2 - 1) + 1, \\
f_1(\mathbf{x}) &= -f_2\big(1.2(\mathbf{x}+2)\big) + 3x_2 - 1.
\end{aligned}
\tag{1}
$$

We can see that between fidelities are nonlinear transformations and non-uniform scaling and shifts. The global maximum is -0.3979 at $(-\pi, 12.275), (\pi, 2.275)$ and $(9.425, 2.475)$.

## 1.2 Park1 Function

The input is four dimensional, $\mathbf{x} = [x_1, x_2, x_3, x_4] \in [0, 1]^4$. We have two fidelities,

$$
\begin{aligned}
f_2(\mathbf{x}) &= \frac{x_1}{2}\left[\sqrt{1 + (x_2 + x_3^2)\frac{x_4}{x_1^2}} - 1\right] + (x_1 + 3x_4)\exp[1 + \sin(x_3)], \\
f_1(\mathbf{x}) &= \left[1 + \frac{\sin(x_1)}{10}\right]f_2(\mathbf{x}) - 2x_1 + x_2^2 + x_3^2 + 0.5.
\end{aligned}
\tag{2}
$$

The global maximum is at 25.5893 at $(1.0, 1.0, 1.0, 1.0)$.

## 1.3 Levy Function

The input is two dimensional, $\mathbf{x} = [x_1, x_2] \in [-10, 10]^2$. The query has three fidelities,

$$
\begin{aligned}
f_3(\mathbf{x}) &= -\sin^2(3\pi x_1) - (x_1 - 1)^2[1 + \sin^2(3\pi x_2)] - (x_2 - 1)^2[1 + \sin^2(2\pi x_2)], \\
f_2(\mathbf{x}) &= -\exp(0.1 \cdot \sqrt{-f_3(\mathbf{x})}) - 0.1 \cdot \sqrt{1 + f_3^2(\mathbf{x})}, \\
f_1(\mathbf{x}) &= -\sqrt{1 + f_3^2(\mathbf{x})}.
\end{aligned}
\tag{3}
$$

The global maximum is 0.0 at $(1.0, 1.0)$.

# 2 Details of Real-World Applications

## 2.1 Mechanical Plate Vibration Design

In this application, we want to make a 3-D simply supported, square, elastic plate, of size $10 \times 10 \times 1$, as shown in Fig. 2. The goal is to find materials that can maximize the fourth vibration mode

frequency (so as to avoid resonance with other parts which causes damages). The materials are parameterized by three properties, Young's modulus (in $[1 \times 10^{11}, 5 \times 10^{11}]$), Poisson's ratio (in $[0.2, 0.6]$) and mass density (in $[6 \times 10^3, 9 \times 10^3]$).

To compute the frequency, we discretize the plate with quadratic tetrahedral elements (see Fig. 2). We consider two fidelities. The low-fidelity solution is obtained from setting a maximum mesh edge length to 1.2, while the high-fidelity 0.6. We then use the finite finite element method (Zienkiewicz et al., 1977) to solve for the first 4th vibration mode and compute the frequency as our objective.

Figure 2: The plate discretized with quadratic tetrahedral elements (the maximum mesh edge length is 1.2).

## 2.2 Thermal Conductor Design

In the second application, we consider the design of a thermal conductor, shown in Fig. 3a. The heat source is on the left, where the temperature is zero at the beginning and ramps to 100 degrees in 0.5 seconds. The heat runs through the conductor to the right end. The size and properties of the conductor are fixed: the thermal conductivity and mass density are both 1. We need to bore a hole in the centre to install the conductor. The edges on the top, bottom and inside the hole are all insulated, *i.e.,* no heat is transferred across these edges. Note that the size and the angle of the hole determine the speed of the heat transfusion. The hole in general is an ellipse, described by three parameters, x-radius, y-radius and angle. The goal is to make the heat conduction (from left to right) as fast as possible. Hence, we use the time to reach 70 degrees on the right end as the objective function value. To compute the time, we discretize the conductor with quadratic tetrahedral elements, and apply the finite element methods to solve a transient heat transfer problem (Incropera et al., 2007) to obtain a response heat curve on the right edge. An example is given in Fig. 3b. The response curve is a function of time, from which we can calculate when the temperature reaches 70 degrees. We consider queries of two fidelities. The low fidelity queries are computed with the maximum mesh edge length being 0.8 in solving the heat transfer problem; the high fidelity queries are computed with the maximum mesh edge length being 0.2.

## 3 Details of Stochastic Variational Learning

We develop a stochastic variational learning algorithm to jointly estimate the posterior of $\mathcal{W} = \{\mathbf{w}_m\}$ — the NN weights in the output layer in each fidelity, and the hyperparameters, including all the other NN weights $\Theta = \{\boldsymbol{\theta}_m\}$ and noise variance $\mathbf{s} = [\sigma_1^2, \ldots, \sigma_M^2]^\top$. To this end, we assume $q(\mathcal{W}) = \prod_{m=1}^M q(\mathbf{w}_m)$ where each $q(\mathbf{w}_m) = \mathcal{N}(\mathbf{w}_m | \boldsymbol{\mu}_m, \boldsymbol{\Sigma}_m)$. We parameterize $\boldsymbol{\Sigma}_m$ with its Cholesky decomposition to ensure the positive definiteness, $\boldsymbol{\Sigma}_m = \mathbf{L}_m \mathbf{L}_m^\top$ where $\mathbf{L}_m$ is a lower triangular matrix. We then construct a variational model evidence lower bound (ELBO) from the joint probability of our model (see (4) of the main paper),

$$\mathcal{L}\big(q(\mathcal{W}), \Theta, \mathbf{s}\big) = \mathbb{E}_q \left[ \frac{\log(p(\mathcal{W}, \mathcal{Y} | \mathcal{X}, \Theta, \mathbf{s})}{q(\mathcal{W})} \right]$$

$$= -\sum_{m=1}^M \text{KL}\big(q(\mathbf{w}_m) \| p(\mathbf{w}_m)\big) + \sum_{m=1}^M \sum_{n=1}^{N_m} \mathbb{E}_q \big[ \log \big( \mathcal{N}(y_{nm} | f_m(\mathbf{x}_{nm}), \sigma_m^2) \big) \big], \quad (4)$$

where $p(\mathbf{w}_m) = \mathcal{N}(\mathbf{w}_m | \mathbf{0}, \mathbf{I})$ and $\text{KL}(\cdot \| \cdot)$ is the Kullback Leibler divergence. We maximize $\mathcal{L}$ to estimate $q(\mathcal{W})$, $\Theta$ and $\mathbf{s}$ jointly. However, since the NN outputs $f_m(\cdot)$ in each fidelity are coupled in a highly nonlinear way (see (3) of the main paper), the expectation terms in $\mathcal{L}$ is analytical intractable.

(a) *Conductor*

(b) *Heat Response Curve*

Figure 3: The thermal conductor with one transient heat solution (a), and the heat responsive curve on the right edge (b). The white triangles in (a) are the finite elements used to discretize the conductor to compute the solution.

To address this issue, we apply stochastic optimization. Specifically, we use the reparameterization trick (Kingma and Welling, 2013) and for each $\mathbf{w}_m$ generate parameterized samples from their variational posterior, $\widehat{\mathbf{w}}_m = \boldsymbol{\mu}_m + \mathbf{L}_m \boldsymbol{\epsilon}$ where $\boldsymbol{\epsilon} \sim \mathcal{N}(\cdot | \mathbf{0}, \mathbf{I})$. We then substitute each sample $\widehat{\mathbf{w}}_m$ for $\mathbf{w}_m$ in computing all $\log \left( \mathcal{N}(y_{nm} | f_m(\mathbf{x}_{nm}), \sigma_m^2) \right)$ in (4) and remove the expectation in front of them. We therefore obtain $\widehat{\mathcal{L}}$, an unbiased estimate of ELBO, which is analytically tractable. Next, we compute $\nabla \widehat{\mathcal{L}}$, which is an unbiased estimate of the $\nabla \mathcal{L}$ and hence can be used to maximize $\mathcal{L}$. We can use any stochastic optimization algorithm.

## 4 Proof of Lemma 4.1

**Lemma 4.1.** *As long as the conditional posterior variance $\gamma(f_{m-1}, \mathbf{x}) > 0$, the posterior variance $\eta_m(\mathbf{x})$, computed based on the quadrature in (7) of the main paper, is positive.*

*Proof.* First, for brevity, we denote $u(t_k, \mathbf{x})$ and $\gamma(t_k, \mathbf{x})$ in (7) of the main paper by $u_k$ and $\gamma_k$, respectively. Then from the quadrature results, we compute the variance

$$\mathrm{Var}(f_m | \mathcal{D}) = \sum_k g_k \gamma_k + \sum_k g_k u_k^2 - (\sum_k g_k u_k)^2.$$

Since $\gamma_k > 0$, the first summation $\sum_k g_k \gamma_k > 0$. Note that the quadrature weights have all $g_k > 0$ and $\sum_k g_k = 1$. We define $\bar{u} = \sum_k g_k u_k$. Next, we derive that

$$
\begin{aligned}
\sum_k g_k u_k^2 - (\sum_k g_k u_k)^2 &= \sum_k g_k u_k^2 - \bar{u}^2 \\
&= \sum_k g_k u_k^2 + \bar{u}^2 - 2\bar{u}^2 \\
&= \sum_k g_k u_k^2 + \sum_k g_k \bar{u}^2 - 2\bar{u}^2 \\
&= \sum_k g_k u_k^2 + \sum_k g_k \bar{u}^2 - 2\sum_k g_k u_k \bar{u} \\
&= \sum_k g_k (u_k^2 + \bar{u}^2 - 2u_k \bar{u}) \\
&= \sum_k g_k (u_k - \bar{u})^2 \geq 0.
\end{aligned}
\tag{5}
$$

Therefore, $\mathrm{Var}(f_m | \mathcal{D}) > 0$. $\qquad \square$

# 5    Proof of Nonnegative Variance in (12) of the Main Paper

We show the variance in (12) of the main paper, computed by quadrature, is non-negative. The proof is very similar to that of Lemma 4.1 (Section 4). We denote the quadrature weights and nodes by $\{g_k\}$ and $\{t_k\}$. Then we have

$$Z = \sum_k g_k R(t_k), \;\; Z_1 = \sum_k g_k t_k R(t_k), \;\;\; Z_2 = \sum_k g_k t_k^2 R(t_k). \tag{6}$$

Therefore,

$$\frac{Z_1}{Z} = \sum_k t_k \frac{g_k R(t_k)}{\sum_j g_j R(t_j)} = \sum_k t_k \nu_k,$$

$$\frac{Z_2}{Z} = \sum_k t_k^2 \frac{g_k R(t_k)}{\sum_j g_j R(t_j)} = \sum_k t_k^2 \nu_k \tag{7}$$

where $\nu_k = \frac{g_k R(t_k)}{\sum_j g_j R(t_j)} > 0$ and $\sum_k \nu_k = 1$. Following the same derivation as in (5), we can immediately show that the variance

$$Z_2/Z - Z_1^2/Z^2 = \sum_k \nu_k (t_k - \bar{t})^2 \geq 0$$

where $\bar{t} = Z_1/Z = \sum_k t_k \nu_k$.