[Reviews · NeurIPS 2020]

Review 1

Summary and Contributions: This paper is about multi-fidelity bayesian optimization. The main contribution is to extend the idea of multi-fidelity max value entropy search from Gaussian processes to neural networks. Computing the max value entropy for neural networks is non trivial. Gauss-Hermite quadrature and variational inference were used to make the computation more tractable. The proposed approach was applied to some synthetic and real problems and claimed to be state of the art.

Strengths: This paper is an useful extension of the GP based MES approach to neural networks. Neural networks have been gaining popularity in Bayesian optimization, and this work addresses an important problem in this domain. Although the individual ideas are not novel, their application to this problem is novel and relevant to the NeurIPS community. The paper provides an involved procedure using Gauss-Hermit quadrature and variational inference to compute the max value entropy. An advantage of the approximation strategy is that it is amenable to gradient based optimizers. The max value entropy is then maximized using L-BFGS to yield the next point to evaluate. The proposed technique seem technically sound.

Weaknesses: The main weakness of this work is the complexity of the approach. Computing the MES acquisition function is not straightforward and requires multiple approximation steps. This probably could not have been avoided though.

Correctness: The presented experimental results raise some skepticism. It is claimed that for one of the synthetic functions, the neural network finds the global optimum after one query point. I believe it is the optimal value rather than the optimal point. How is this significant and why does is imply that the neural network is better at estimating the unknown function? Another concern is about the choice of hyper-parameters for the neural network. SMAC3 was used to tune the hyper-parameters of the network on some initial dataset. In practice however one will not have such liberty to choose hyper-parameters on some held out dataset. More details must be provided about the hyper-parameter selection process and the dataset used. Also, what objective was maximized while optimizing the hyper-parameters? There is a potential risk of overfitting if the hyper-parameters were optimized on the same tasks.

Clarity: Most of the paper is very clearly written. The methods section is a bit dense, but understandable on careful reading. On the other hand, details are missing from the experiments section.

Relation to Prior Work: Yes related work seems exhaustive to me.

Reproducibility: No

Additional Feedback: Overall this is a nice idea. I will increase my score if the authors address my concerns about the experiments. ========== After rebuttal: I am satisfied by the explanation for choosing the hyper-parameters. However, I also partially agree on the slight lack of novelty, so I will keep my earlier score. Since this is mainly an experimental paper, the experiments should be further improved based on the reviews. Also, since the method is heavily dependent on the approximations and hyper-parameters, the implementation details should be provided or the code should be released.


Review 2

Summary and Contributions: This paper proposes a novel method for multi-fidelity Bayesian optimization. In contrast to previous methods which mostly rely on Gaussian process models and which typically can't capture the complex correlations between function outputs at different fidelities, the authors propose to use deep neural network based models (one for each fidelity) that are flexible enough to capture such correlations. Following the popular DNGO method, the paper proposes to keep all but the last layer of each neural network fixed as hyperparameters, and perform Bayesian inference only over the last layers. To this end, the authors use a stochastic variational inference method. In order to approximate the (conditional) posteriors over the outputs for each fidelity, the method employs a combination of Gauss-Hermite quadrature and moment matching. The posterior is then used to compute and optimize an information-based acquisition function (i.e. max-value entropy search) in a tractable way. Experiments on synthetic and real-world benchmarks demonstrate that the proposed DNN-MFBO method significantly outperforms previous multi-fidelty BO methods at smaller query cost.

Strengths: This paper tackles the important problem of multi-fidelity Bayesian optimization, and proposes a novel, principled, efficient and effective method, combining deep neural network based surrogate models with max-value entropy search. The methodology seems sound, and the empirical results are impressive. The paper is very relevant to the NeurIPS community, and it makes significant contributions that will be of interest to many researchers and practitioners in machine learning.

Weaknesses: The idea of using a neural network based surrogate model in the context of Bayesian optimization was first proposed by Snoek et al. 2015 in a popular method called DNGO. Furthermore, neural network models were used for the closely-related multi-task Bayesian optimization setting in Perrone et al. 2018, which the authors fail to cite and do not seem to be aware of. While Perrone et al. 2018 consider the particular application of hyperparameter tuning of machine learning models, their approach seems general enough to be applied to other problems as well. Given that multi-fidelity BO can be viewed as a special case of multi-task BO, it would be important to contrast the work of Perrone et al. 2018 with the method proposed in this paper. The proposed method seems rather complicated and involved, with several layers of approximation to make things tractable. As a result, I am not overly confident that it would be straightforward to implement the algorithm and reproduce the presented results. I thus strongly encourage the authors to provide code for their method. Furthermore, the benchmark functions used in the empirical evaluation are somewhat small-scale, each only involving a couple of parameters to optimize. It would make the paper significantly stronger if it would present results on more challenging, higher-dimensional optimization tasks. For example, an important application domain of Bayesian optimization (especially in the multi-fidelity context) is hyperparameter tuning of machine learning algorithms, which I would really like to see results on. References: Perrone et al., "Scalable Hyperparameter Transfer Learning", NeurIPS 2018

Correctness: Yes, the claims, derivations and empirical methodology appear correct, as far as I can tell.

Clarity: All-in-all, the paper is fairly well written and somewhat easy to follow. That being said, Section 4 is a bit dense and contains a lot of notation, which makes it a bit harder to read. The pseudocodes presented in Algorithms 1 and 2 are helpful to get a high-level overview over the algorithm.

Relation to Prior Work: Prior work and its relation to the proposed method is discussed extensively, mostly in Section 5 (which is quite long for a related work section); all relevant literature appears to be cited and discussed, apart from Perrone et al. 2018, as pointed out in "weaknesses" above.

Reproducibility: No

Additional Feedback: POST-REBUTTAL: Thank you for addressing some of my concerns. I am still very keen on seeing larger scale experiments, but appreciate the novelty and technical methodology, which will be useful to the community. Overall, my sentiment of the paper has not changed and I am keeping my score at 6 -- I am still in favour of seeing it accepted, although I am not overly enthusiastic due to the concerns mentioned. In any case, I strongly encourage the authors to continue working on what seems to be a very promising research direction, and to take into account all feedback in order to improve their work. ============== Questions: - in the experiments, why did you use different kernels for the different competing methods? does this provide a fair comparison? also, what's the difference between the SE kernel used in SF-MES and MF-GP-UCB and the RBF kernel used in MF-MES and MF-SKO? - in the plots, why do the methods not start at the same regret? aren't all methods using the same initial training points? or do the plots not show the initial training points? if so, I think the plots should be changed to include all queried points, to convince the reader that the evaluation was fair - how do your modeling choices contrast to those made in Perrone et al. 2018, where a shared neural network trunk is used to learn features shared across different tasks, with different heads for the different tasks; why do you think that your approach of having independent neural networks that are sequentially connected via their outputs/inputs is more suitable for the multi-fidelity scenario? - have you considered acquisition functions other than max-value entropy search? why did you decide to use that particular one? could your approach be straightforwardly extended to other acquisition functions? Minor: - the authors seem to have modified the style file/spacing; e.g., the margin before section titles appears too small; I strongly encourage the authors to stick to the style file; to save space, the authors could e.g. move the details on the experimental settings (i.e., in the third paragraph of Section 6.1) to the appendix, which is quite long and not so critical - Algorithm 1: the spacing after the last line seems a bit small - l. 56: "in three" --> "on three" - l. 57: "engineer design" --> "engineering design" - l. 182: typo "maximums" - l. 263: "in three" --> "on three"


Review 3

Summary and Contributions: [Please find the updates in the "Additional feedback" section.] The paper proposes a multi-fidelity model based on neural networks to perform Bayesian optimization. The model uses a neural network to model each data source and then stacks them according to fidelity level. Gauss-Hermite quadrature and moment matching are used to approximate analytically intractable posteriors. Monte-Carlo method is used to approximate the acquisition function.

Strengths: The work is well structured and has detailed technical derivations of the key components. According to experiments the proposed method outperforms SOTA.

Weaknesses: Lack of experiments and ablation study (how do acquisition functions and network architectures affect the convergence?). The work is ok, but incremental: the idea of multifidelity model based on neural networks is not novel (see e.g. this work https://arxiv.org/abs/1903.00104); the aquisition function and its calculation is borrowed from the work of Wang, Z. and Jegelka, S. (2017). Max-value entropy search for efficient bayesian optimization.

Correctness: The overall methodology is correct, but I have not checked the correctness of mathematical derivations details. Some of the experiments look suspicious: why does not DNN-MFBO start with the same regret as other methods in figure 1.b and 1.f?

Clarity: Yes

Relation to Prior Work: Prior art about multi-fidelity deep learning models is not reviewed.

Reproducibility: No

Additional Feedback: Update after reading the rebuttal: "It is baseless and unfair to claim that our work has NO novelty just because of some lightly related and essentially different work." > to my mind, changing the objective or inference method hardly makes the multi-fidelity neural network essentially different, unless the objective or method is indeed novel, but it is not the case of this work. "The reviewer missed our key contribution and claimed our calculation is just borrowed from (Wang, et. al. 2017)" > as far as I understood the authors just replaced MCMC with another well-known approximation, that has its own advantages and disadvantages. Thus, despite the work is technically fine, it doesn't bring scientific insights, that one would anticipate from a NeurIPS paper, especially along with absense of ablation study. What if the results achieved by the proposed method are good merely because of the neural network model that estimates the objective, and the criterion doesn't matter much?


Review 4

Summary and Contributions: In order to capture the strong and complex correlations across the multi-fidelity data, this paper presents a multi-fidelity Bayesian neural network model, and then integrates it into the multi-fidelity Bayesian optimization. Algorithm performance has been verified on various cases.

Strengths: By augmenting the inputs with outputs from lower fidelities, the presented multi-fidelity Bayesian neural network is enabled to capture complicated correlations. Also, the fidelity-wise Gauss-Hermite quadrature and moment-matching have been studied in order to calculate the mutual information based acquisition function required in BO.

Weaknesses: The novelty of this paper is incremental, since multi-fidelity modeling structure has been investigated in the following paper: Cutajar, K., Pullin, M., Damianou, A., Lawrence, N., & González, J. (2019). Deep Gaussian processes for multi-fidelity modeling. arXiv preprint arXiv:1903.07320. The main difference is that Cutajar et al (2019) used the deep Gaussian process for modeling while the authors herein employed the Bayesian NN.

Correctness: The claims and method in this paper sound good.

Clarity: The paper is easy to follow and understand.

Relation to Prior Work: The difference has been clearly discussed.

Reproducibility: No

Additional Feedback: (1) The difference to the multi-fidelity modeling proposed by Cutajar et al (2019) should be made clear for highlighting the novelty of this work; (2) It seems that the improvement of the proposed algorithm is fully brought by the powerful multi-fidelity Bayesian neural networks. Hence, the paper focuses more on a multi-fidelity modeling method rather than a new multi-fidelity optimization framework. Along this line, the comparison seems to be unfair since other multi-fidelity Bayesian optimization algorithms use only for example the kriging or GP model.

[Author Response · NeurIPS 2020]

**To Reviewer 1:**

R1: We do agree that our approach is complex and involves "multiple approximation steps". However, we view it as a contribution rather than "weakness", because it enables multi-fidelity optimization with a highly expressive DNN model. The experiments have shown not only better optimization performance, but also much higher computational efficiency (Fig. 2).

C2: On *Park1*, the NN "finds the global optimum after one query point"— How is this significant... "

R2: Great question. First, the quality of the query point very much depends on the accuracy of the surrogate model. The first query of our NN (trained with initial points) has already been very close to the optimum point (Fig. 1b), showing its superior quality in approximating the objective (contrast to MF-MES using the same acquisition function). Second, adding one more point, our NN coincides with the objective in the optimum value (Fig. 1f), confirming its capability of approximating/estimating the objective.

C3: Details about hyper-parameter selection; no liberty to choose a heldout dataset in practice.

R3: We did not use heldout datasets. We randomly split the initial data points multiple times, each time with half for training and the remaining half for test. We optimized the hyper-parameters to minimize the average test error. We will supplement these details.

**To Reviewer 2:** Thanks for suggesting us to test high-dimensional optimization problems. We will do it. Extending our approach to other acquisition functions is not straightforward and we have included it in our future research plan. We will release our code.

C4: Compare with multi-task BO in (Perrone et al. 2018); contrast to the multi-task modeling choice

R4: Thanks for providing this great reference. We will cite and compare with it. Note that we have compared with SOTA MF-BO based on multi-task GPs (MF-MES). The chain structure of our model can flexibly capture the strong, complex correlations (*e.g.,* nonlinear, nonstationary) between successive fidelities, so as to enhance the function approximation in each (higher) fidelity. The multi-task model in (Perrone et al. 2018) views each fidelity (task) as symmetric and does not reflect the monotonicity of function accuracy/importance along with the fidelities. More important, it does not model the correlation between fidelities — given the shared bases, different fidelities are assumed to be independent. Note that MF-MES assumes a linear fidelity correlation and has been worse than our method, implying that accounting for the complex relationships of the fidelities is necessary and beneficial.

C5: Why use different kernels for competing methods? What is the difference between SE, ARD and RBF?

R5: Great question. For all the competing methods, we used the original implementations and settings that give the best results reported in their papers. The used kernels, however, have minor differences. SE is RBF multiplied by a coefficient (called "amplitude"); ARD introduces a length scale parameter for each input dimension while RBF uses one for all.

C6: Why do not the methods start with the same regret? aren't all methods using the same initial training points...

R6: The initial training sets are the same for all the methods. We started to report the regrets after the first query. These methods learn different models to approximate the objective, and use different procedures to calculate the acquisition function and select the query points. As a result, their regrets are unlikely to be identical, even at the beginning.

**To Reviewer 3:** Regarding the initial regrets, please see R6.

C7: Lack of ablation study

R7: We followed the paper of SOTA MF-BO (MF-MES) to perform comprehensive experiments in both synthetic and real datasets. We followed the DNN-BO by Snoek et. al. 2015 to use another BO to select the hyper-parameters, and to evaluate the computational efficiency. Hence, we argue that our experiments are enough. However, we agree that more studies will be helpful and will add more.

C8: The work is not novel because the multi-fidelity model (Meng & Karniadakis, 2019) was proposed and the acquisition function and calculation are borrowed from (Wang et. al. 2017).

R8: First, we agree that our MF model structure resembles (Meng & Karniadakis, 2019) in some degree (though the latter still has quite a few critically distinct components, *e.g.,* differential NN and linear correlations). We would love to cite and discuss about it. However, the two models are totally different in the goal (integrating PDEs and multi-fidelity data for prediction *vs.* optimizing blackbox functions with multi-fidelity queries), training objective (summation of several heuristic square losses *vs.* a principled variational evidence bound), and inference (pure point estimations *vs.* Bayesian inference with uncertainty quantification/propagation). It is **baseless** and **unfair** to claim that our work has NO novelty just because of some lightly related and essentially different work.

Second and more important, the critical contribution of our work is that we use quadrature and moment-matching to develop a highly efficient and tractable approach to calculate and optimize the widely used MES acquisition function for DNNs (Sec. 4). This task is very challenging and our solution is never covered by the prior work. The reviewer missed our key contribution and claimed our calculation is just borrowed from (Wang, et. al. 2017) (applicable to a single-fidelity GP only), which is **wrong** and **irrational**. Following the reviewer's reasoning, the many excellent BO works (see Sec.5 Related Work) that use MES principle should all be judged as "not novel" and "incremental".

**To Reviewer 4:**

C9: The model is not novel due to the deep GP mode in (Cutajar et. al., 2019); "the paper focuses more on a multi-fidelity modeling rather than multi-fidelity optimization framework"; "the comparison seems to be unfair since other MFBO use only GP models"

R9: We are astonished by these **baseless** and **erroneous** conclusions! First, despite some high-level similarity, our model and (Cutajar et. al. 2019) belong to distinct families: DNN and GP. The goals and inferences are totally different. Following your reasoning, deep GP should be viewed as not novel because it plagiarizes the idea of "deep" architectures in DNNs! Second, the dominant space of our paper is used to introduce our efficient and tractable approach to compute the acquisition function for multi-fidelity optimization (page 4-6). This is our major contribution (see R1&R9). The multi-fidelity modeling only takes half a page (page 3). How do you conclude our work " focuses more on a multi-fidelity modeling rather than multi-fidelity optimization framework"? Third, we compared with GP based methods, because the SOTA MFBO are all based on GPs! This is for a fair comparison. We will compare with the multi-task model suggested by Reviewer 2. However, preserving results of the GP based MFBO is necessary to ensure the fairness and comprehensiveness.

[Meta-Review · NeurIPS 2020]

*PROS: extension of multi-fidelity max value entropy search from Gaussian processes to neural networks *CONS: there are concern is about the choice of hyper-parameters for the neural network, The proposed method seems rather complicated and involved with concerns for reproducibility, small scale evaluation Meta-reviewer recommendations: The paper is clearly borderline with R1 and R2 voting for acceptance and R3 and R4 leaning towards rejection. R3 seems overly negative in a not very well justified manner. R4's review is very short and mainly indicates that the difference to the multi-fidelity modeling proposed by Cutajar et al (2019) should be made clear. I believe the authors successfully address R4's comments in the rebuttal. I recommend the authors to perform an ablation study recommended by R3 in their final version. As recommended, the experiments should be further improved based on the reviews. The method seems heavily dependent on the approximations and hyper-parameters, the implementation details should be provided or the code should be released for reproducibility.